# Decomposition Bounds for Marginal MAP

**Wei Ping**[*]       **Qiang Liu**[†]       **Alexander Ihler**[*]
[*]Computer Science, UC Irvine    [†]Computer Science, Dartmouth College
{wping,ihler}@ics.uci.edu   qliu@cs.dartmouth.edu

## Abstract

Marginal MAP inference involves making MAP predictions in systems defined with latent variables or missing information. It is significantly more difficult than pure marginalization and MAP tasks, for which a large class of efficient and convergent variational algorithms, such as dual decomposition, exist. In this work, we generalize dual decomposition to a generic *power sum* inference task, which includes marginal MAP, along with pure marginalization and MAP, as special cases. Our method is based on a block coordinate descent algorithm on a new convex decomposition bound, that is guaranteed to converge monotonically, and can be parallelized efficiently. We demonstrate our approach on marginal MAP queries defined on real-world problems from the UAI approximate inference challenge, showing that our framework is faster and more reliable than previous methods.

## 1   Introduction

Probabilistic graphical models such as Bayesian networks and Markov random fields provide a useful framework and powerful tools for machine learning. Given a graphical model, *inference* refers to answering probabilistic queries about the model. There are three common types of inference tasks. The first are max-inference or maximum a *posteriori* (MAP) tasks, which aim to find the most probable state of the joint probability; exact and approximate MAP inference is widely used in structured prediction [26]. Sum-inference tasks include calculating marginal probabilities and the normalization constant of the distribution, and play a central role in many learning tasks (e.g., maximum likelihood). Finally, marginal MAP tasks are "mixed" inference problems, which generalize the first two types by marginalizing a subset of variables (e.g., hidden variables) before optimizing over the remainder.[1] These tasks arise in latent variable models [e.g., 29, 25] and many decision-making problems [e.g., 13]. All three inference types are generally intractable; as a result, approximate inference, particularly convex relaxations or upper bounding methods, are of great interest.

Decomposition methods provide a useful and computationally efficient class of bounds on inference problems. For example, dual decomposition methods for MAP [e.g., 31] give a class of easy-to-evaluate upper bounds which can be directly optimized using coordinate descent [36, 6], subgradient updates [14], or other methods [e.g., 22]. It is easy to ensure both convergence, and that the objective is monotonically decreasing (so that more computation always provides a better bound). The resulting bounds can be used either as stand-alone approximation methods [6, 14], or as a component of search [11]. In summation problems, a notable decomposition bound is tree-reweighted BP (TRW), which bounds the partition function with a combination of trees [e.g., 34, 21, 12, 3]. These bounds are useful in joint inference and learning (or "inferning") frameworks, allowing learning with approximate inference to be framed as a joint optimization over the model parameters and decomposition bound, often leading to more efficient learning [e.g., 23]. However, far fewer methods have been developed for marginal MAP problems.

In this work, we deveop a decomposition bound that has a number of desirable properties: (1) *Generality*: our bound is sufficiently general to be applied easily to marginal MAP. (2) *Any-time*: it yields a bound at any point during the optimization (not just at convergence), so it can be used in an any-time way. (3) *Monotonic and convergent*: more computational effort gives strictly tighter bounds; note that (2) and (3) are particularly important for high-width approximations, which are expensive to represent and update. (4) Allows *optimization over all parameters*, including the "weights", or fractional counting numbers, of the approximation; these parameters often have a significant effect on the tightness of the resulting bound. (5) *Compact representation*: within a given class of bounds, using fewer parameters to express the bound reduces memory and typically speeds up optimization.

We organize the rest of the paper as follows. Section 2 gives some background and notation, followed by connections to related work in Section 3. We derive our decomposed bound in Section 4, and present a (block) coordinate descent algorithm for monotonically tightening it in Section 5. We report experimental results in Section 6 and conclude the paper in Section 7.

## 2 Background

Here, we review some background on graphical models and inference tasks. A Markov random field (MRF) on discrete random variables $x = [x_1, \ldots, x_n] \in \mathcal{X}^n$ is a probability distribution,

$$p(x; \theta) = \exp\Big[\sum_{\alpha \in \mathcal{F}} \theta_\alpha(x_\alpha) - \Phi(\theta)\Big]; \quad \Phi(\theta) = \log\sum_{x \in \mathcal{X}^n} \exp\Big[\sum_{\alpha \in \mathcal{F}} \theta_\alpha(x_\alpha)\Big], \quad (1)$$

where $\mathcal{F}$ is a set of subsets of the variables, each associated with a factor $\theta_\alpha$, and $\Phi(\theta)$ is the log partition function. We associate an undirected graph $G = (V, E)$ with $p(x)$ by mapping each $x_i$ to a node $i \in V$, and adding an edge $ij \in E$ iff there exists $\alpha \in \mathcal{F}$ such that $\{i, j\} \subseteq \alpha$. We say node $i$ and $j$ are neighbors if $ij \in E$. Then, $\mathcal{F}$ is the subset of cliques (fully connected subgraphs) of $G$.

The use and evaluation of a given MRF often involves different types of inference tasks. *Marginalization*, or *sum-inference* tasks perform a sum over the configurations to calculate the log partition function $\Phi$ in (1), marginal probabilities, or the probability of some observed evidence. On the other hand, the maximum *a posteriori* (MAP), or *max-inference* tasks perform joint maximization to find configurations with the highest probability, that is, $\Phi_0(\theta) = \max_x \sum_{\alpha \in \mathcal{F}} \theta_\alpha(x_\alpha)$.

A generalization of max- and sum- inference is *marginal MAP*, or *mixed-inference*, in which we are interested in first marginalizing a subset $A$ of variables (e.g., hidden variables), and then maximizing the remaining variables $B$ (whose values are of direct interest), that is,

$$\Phi_{AB}(\theta) = \max_{x_B} Q(x_B) = \max_{x_B} \log\sum_{x_A} \exp\Big[\sum_{\alpha \in \mathcal{F}} \theta_\alpha(x_\alpha)\Big], \quad (2)$$

where $A \cup B = V$ (all the variables) and $A \cap B = \emptyset$. Obviously, both sum- and max- inference are special cases of marginal MAP when $A = V$ and $B = V$, respectively.

It will be useful to define an even more general inference task, based on a power sum operator:

$$\sum_{x_i}^{\tau_i} f(x_i) = \Big[\sum_{x_i} f(x_i)^{1/\tau_i}\Big]^{\tau_i},$$

where $f(x_i)$ is any non-negative function and $\tau_i$ is a *temperature* or *weight* parameter. The power sum reduces to a standard sum when $\tau_i = 1$, and approaches $\max_x f(x)$ when $\tau_i \to 0^+$, so that we define the power sum with $\tau_i = 0$ to equal the max operator.

The power sum is helpful for unifying max- and sum- inference [e.g., 35], as well as marginal MAP [15]. Specifically, we can apply power sums with different weights $\tau_i$ to each variable $x_i$ along a predefined elimination order (e.g., $[x_1, \ldots, x_n]$), to define the *weighted log partition function*:

$$\Phi_{\boldsymbol{\tau}}(\theta) = \log\sum_{x}^{\boldsymbol{\tau}} \exp(\theta(x)) = \log\sum_{x_n}^{\tau_n} \ldots \sum_{x_1}^{\tau_1} \exp(\theta(x)), \quad (3)$$

where we note that the value of (3) depends on the elimination order unless all the weights are equal. Obviously, (3) includes marginal MAP (2) as a special case by setting weights $\tau_A = 1$ and $\tau_B = 0$. This representation provides a useful tool for understanding and deriving new algorithms for general inference tasks, especially marginal MAP, for which relatively few efficient algorithms exist.

# 3   Related Work

Variational upper bounds on MAP and the partition function, along with algorithms for providing fast, convergent optimization, have been widely studied in the last decade. In MAP, dual decomposition and linear programming methods have become a dominating approach, with numerous optimization techniques [36, 6, 32, 14, 37, 30, 22], and methods to tighten the approximations [33, 14].

For summation problems, most upper bounds are derived from the tree-reweighted (TRW) family of convex bounds [34], or more generally conditional entropy decompositions [5]. TRW bounds can be framed as optimizing over a convex combination of tree-structured models, or in a dual representation as a message-passing, TRW belief propagation algorithm. This illustrates a basic tension in the resulting bounds: in its primal form [2] (combination of trees), TRW is inefficient: it maintains a weight and $O(|V|)$ parameters for each tree, and a large number of trees may be required to obtain a tight bound; this uses memory and makes optimization slow. On the other hand, the dual, or free energy, form uses only $O(|E|)$ parameters (the TRW messages) to optimize over the set of all possible spanning trees – but, the resulting optimization is only guaranteed to be a bound at convergence, [3] making it difficult to use in an anytime fashion. Similarly, the gradient of the weights is only correct at convergence, making it difficult to optimize over these parameters; most implementations [e.g., 24] simply adopt fixed weights.

Thus, most algorithms do not satisfy all the desirable properties listed in the introduction. For example, many works have developed convergent message-passing algorithms for convex free energies [e.g., 9, 10]. However, by optimizing the dual they do not provide a bound until convergence, and the representation and constraints on the counting numbers do not facilitate optimizing the bound over these parameters. To optimize counting numbers, [8] adopt a more restrictive free energy form requiring positive counting numbers on the entropies; but this cannot represent marginal MAP, whose free energy involves conditional entropies (equivalent to the difference between two entropy terms).

On the other hand, working in the primal domain ensures a bound, but usually at the cost of enumerating a large number of trees. [12] heuristically select a small number of trees to avoid being too inefficient, while [21] focus on trying to speed up the updates on a given collection of trees. Another primal bound is weighted mini-bucket (WMB, [16]), which can represent a large collection of trees compactly and is easily applied to marginal MAP using the weighted log partition function viewpoint [15, 18]; however, existing optimization algorithms for WMB are non-monotonic, and often fail to converge, especially on marginal MAP tasks.

While our focus is on variational bounds [16, 17], there are many non-variational approaches for marginal MAP as well. [27, 38] provide upper bounds on marginal MAP by reordering the order in which variables are eliminated, and using exact inference in the reordered join-tree; however, this is exponential in the size of the (unconstrained) treewidth, and can easily become intractable. [20] give an approximation closely related to mini-bucket [2] to bound the marginal MAP; however, unlike (weighted) mini-bucket, these bounds cannot be improved iteratively. The same is true for the algorithm of [19], which also has a strong dependence on treewidth. Other examples of marginal MAP algorithms include local search [e.g., 28] and Markov chain Monte Carlo methods [e.g., 4, 39].

# 4   Fully Decomposed Upper Bound

In this section, we develop a new general form of upper bound and provide an efficient, monotonically convergent optimization algorithm. Our new bound is based on fully decomposing the graph into disconnected cliques, allowing very efficient local computation, but can still be as tight as WMB or the TRW bound with a large collection of spanning trees once the weights and shifting variables are chosen or optimized properly. Our bound reduces to dual decomposition for MAP inference, but is applicable to more general mixed-inference settings.

Our main result is based on the following generalization of the classical Hölder's inequality [7]:

**Theorem 4.1.** *For a given graphical model $p(x; \theta)$ in (1) with cliques $\mathcal{F} = \{\alpha\}$ and a set of non-negative weights $\boldsymbol{\tau} = \{\tau_i \geq 0, i \in V\}$, we define a set of "split weights" $\mathbf{w}^\alpha = \{w_i^\alpha \geq 0, i \in \alpha\}$ on each variable-clique pair $(i, \alpha)$, that satisfies $\sum_{\alpha|\alpha \ni i} w_i^\alpha = \tau_i$. Then we have*

$$\sum_x^{\boldsymbol{\tau}} \prod_{\alpha \in \mathcal{F}} \exp\left[\theta_\alpha(x_\alpha)\right] \leq \prod_{\alpha \in \mathcal{F}} \sum_{x_\alpha}^{\mathbf{w}^\alpha} \exp\left[\theta_\alpha(x_\alpha)\right], \tag{4}$$

*where the left-hand side is the powered-sum along order $[x_1, \ldots, x_n]$ as defined in (3), and the right-hand side is the product of the powered-sums on subvector $x_\alpha$ with weights $\mathbf{w}^\alpha$ along the same elimination order; that is, $\sum_{x_\alpha}^{\mathbf{w}^\alpha} \exp\left[\theta_\alpha(x_\alpha)\right] = \sum_{x_{k_c}}^{w_{k_c}^\alpha} \cdots \sum_{x_{k_1}}^{w_{k_1}^\alpha} \exp\left[\theta_\alpha(x_\alpha)\right]$, where $x_\alpha = [x_{k_1}, \ldots, x_{k_c}]$ should be ranked with increasing index, consisting with the elimination order $[x_1, \ldots, x_n]$ as used in the left-hand side.*

Proof details can be found in Section E of the supplement. A key advantage of the bound (4) is that it decomposes the joint power sum on $x$ into a product of independent power sums over smaller cliques $x_\alpha$, which significantly reduces computational complexity and enables parallel computation.

## 4.1 Including Cost-shifting Variables

In order to increase the flexibility of the upper bound, we introduce a set of *cost-shifting* or *reparameterization* variables $\delta = \{\delta_i^\alpha(x_i) \mid \forall(i, \alpha), i \in \alpha\}$ on each variable-factor pair $(i, \alpha)$, which can be optimized to provide a much tighter upper bound. Note that $\Phi_{\boldsymbol{\tau}}(\theta)$ can be rewritten as,

$$\Phi_{\boldsymbol{\tau}}(\theta) = \log \sum_x^{\boldsymbol{\tau}} \exp\left[\sum_{i \in V} \sum_{\alpha \in N_i} \delta_i^\alpha(x_i) + \sum_{\alpha \in \mathcal{F}} \left(\theta_\alpha(x_\alpha) - \sum_{i \in \alpha} \delta_i^\alpha(x_i)\right)\right],$$

where $N_i = \{\alpha \mid \alpha \ni i\}$ is the set of cliques incident to $i$. Applying inequality (4), we have that

$$\Phi_{\boldsymbol{\tau}}(\theta) \leq \sum_{i \in V} \log \sum_{x_i}^{w_i} \exp\left[\sum_{\alpha \in N_i} \delta_i^\alpha(x_i)\right] + \sum_{\alpha \in \mathcal{F}} \log \sum_{x_\alpha}^{\mathbf{w}^\alpha} \exp\left[\theta_\alpha(x_\alpha) - \sum_{i \in \alpha} \delta_i^\alpha(x_i)\right] \stackrel{\text{def}}{=\!=} L(\delta, \mathbf{w}), \tag{5}$$

where the nodes $i \in V$ are also treated as cliques within inequality (4), and a new weight $w_i$ is introduced on each variable $i$; the new weights $\mathbf{w} = \{w_i, w_i^\alpha \mid \forall(i, \alpha), i \in \alpha\}$ should satisfy

$$w_i + \sum_{\alpha \in N_i} w_i^\alpha = \tau_i, \quad w_i \geq 0, \quad w_i^\alpha \geq 0, \quad \forall(i, \alpha). \tag{6}$$

The bound $L(\delta, \mathbf{w})$ is convex w.r.t. the cost-shifting variables $\delta$ and weights $\mathbf{w}$, enabling an efficient optimization algorithm that we present in Section 5. As we will discuss in Section 5.1, these shifting variables correspond to Lagrange multipliers that enforce a moment matching condition.

## 4.2 Dual Form and Connection With Existing Bounds

It is straightforward to see that our bound in (5) reduces to dual decomposition [31] when applied on MAP inference with all $\tau_i = 0$, and hence $w_i = w_i^\alpha = 0$. On the other hand, its connection with sum-inference bounds such as WMB and TRW is seen more clearly via a dual representation of (5):

**Theorem 4.2.** *The tightest upper bound obtainable by (5), that is,*

$$\min_{\mathbf{w}} \min_{\delta} L(\delta, \mathbf{w}) = \min_{\mathbf{w}} \max_{\mathbf{b} \in \mathbb{L}(G)} \left\{ \langle \theta, b \rangle + \sum_{i \in V} w_i H(x_i; b_i) + \sum_{\alpha \in \mathcal{F}} \sum_{i \in \alpha} w_i^\alpha H(x_i | x_{\mathrm{pa}_i^\alpha}; b_\alpha) \right\}, \tag{7}$$

*where $\mathbf{b} = \{b_i(x_i), b_\alpha(x_\alpha) \mid \forall(i, \alpha), i \in \alpha\}$ is a set of pseudo-marginals (or beliefs) defined on the singleton variables and the cliques, and $\mathbb{L}$ is the corresponding local consistency polytope defined by $\mathbb{L}(G) = \{\mathbf{b} \mid b_i(x_i) = \sum_{x_{\alpha \setminus i}} b_\alpha(x_\alpha), \sum_{x_i} b_i(x_i) = 1\}$. Here, $H(\cdot)$ are their corresponding marginal or conditional entropies, and $\mathrm{pa}_i^\alpha$ is the set of variables in $\alpha$ that rank later than $i$, that is, for the global elimination order $[x_1, \ldots, x_n]$, $\mathrm{pa}_i^\alpha = \{j \in \alpha \mid j \succ i\}$.*

The proof details can be found in Section F of the supplement. It is useful to compare Theorem 4.2 with other dual representations. As the sum of non-negatively weighted conditional entropies, the bound is clearly convex and within the general class of conditional entropy decompositions (CED) [5], but unlike generic CED it has a simple and efficient primal form (5). [4] Comparing

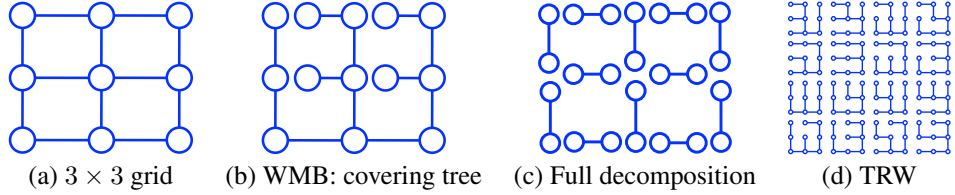

| (a) $3 \times 3$ grid | (b) WMB: covering tree | (c) Full decomposition | (d) TRW |

Figure 1: Illustrating WMB, TRW and our bound on (a) $3 \times 3$ grid. (b) WMB uses a covering tree with a minimal number of splits and cost-shifting. (c) Our decomposition (5) further splits the graph into small cliques (here, edges), introducing additional cost-shifting variables but allowing for easier, monotonic optimization. (d) Primal TRW splits the graph into many spanning trees, requiring even more cost-shifting variables. Note that all three bounds attain the same tightness after optimization.

to the dual form of WMB in Theorem 4.2 of [16], our bound is as tight as WMB, and hence the class of TRW / CED bounds attainable by WMB [16]. Most duality-based forms [e.g., 9, 10] are expressed in terms of joint entropies, $\langle \theta, b \rangle + \sum_\beta c_\beta H(b_\beta)$, rather than conditional entropies; while the two can be converted, the resulting counting numbers $c_\beta$ will be differences of weights $\{w_i^\alpha\}$, [5] which obfuscates its convexity, makes it harder to maintain the relative constraints on the counting numbers during optimization, and makes some counting numbers negative (rendering some methods inapplicable [8]). Finally, like most variational bounds in dual form, the RHS of (7) has a inner maximization and hence guaranteed to bound $\Phi_\tau(\theta)$ only at its optimum.

In contrast, our Eq. (5) is a primal bound (hence, a bound for any $\delta$). It is similar to the primal form of TRW, except that (1) the individual regions are single cliques, rather than spanning trees of the graph, [6] and (2) the fraction weights $\mathbf{w}^\alpha$ associated with each region are vectors, rather than a single scalar. The representation's efficiency can be seen with an example in Figure 1, which shows a $3 \times 3$ grid model and three relaxations that achieve the same bound. Assuming $d$ states per variable and ignoring the equality constraints, our decomposition in Figure 1(c) uses $24d$ cost-shifting parameters ($\delta$), and 24 weights. WMB (Figure 1(b)) is slightly more efficient, with only $8d$ parameters for $\delta$ and and 8 weights, but its lack of decomposition makes parallel and monotonic updates difficult. On the other hand, the equivalent primal TRW uses 16 spanning trees, shown in Figure 1(d), for $16 \cdot 8 \cdot d^2$ parameters, and 16 weights. The increased dimensionality of the optimization slows convergence, and updates are non-local, requiring full message-passing sweeps on the involved trees (although this cost can be amortized in some cases [21]).

## 5 Monotonically Tightening the Bound

In this section, we propose a block coordinate descent algorithm (Algorithm 1) to minimize the upper bound $L(\delta, \mathbf{w})$ in (5) w.r.t. the shifting variables $\delta$ and weights $\mathbf{w}$. Our algorithm has a monotonic convergence property, and allows efficient, distributable local computation due to the full decomposition of our bound. Our framework allows generic powered-sum inference, including max-, sum-, or mixed-inference as special cases by setting different weights.

### 5.1 Moment Matching and Entropy Matching

We start with deriving the gradient of $L(\delta, \mathbf{w})$ w.r.t. $\delta$ and $\mathbf{w}$. We show that the zero-gradient equation w.r.t. $\delta$ has a simple form of moment matching that enforces a consistency between the *singleton beliefs* with their related *clique beliefs*, and that of weights $\mathbf{w}$ enforces a consistency of *marginal* and *conditional entropies*.

**Theorem 5.1.** *(1) For $L(\delta, \mathbf{w})$ in (5), its zero-gradient w.r.t. $\delta_i^\alpha(x_i)$ is*

$$\frac{\partial L}{\partial \delta_i^\alpha(x_i)} = \mu_i(x_i) - \sum_{x_{\alpha \setminus i}} \mu_\alpha(x_\alpha) = 0, \tag{8}$$

**Algorithm 1** Generalized Dual-decomposition (GDD)
---
**Input:** weights $\{\tau_i \mid i \in V\}$, elimination order $\mathbf{o}$.
**Output:** the optimal $\delta^*, \mathbf{w}^*$ giving tightest upper bound $\mathrm{L}(\delta^*, \mathbf{w}^*)$ for $\Phi_\tau(\theta)$ in (5).

initialize $\delta = 0$ and weights $\mathbf{w} = \{w_i, w_i^\alpha\}$.
**repeat**
  **for** node i (in parallel with node j, $(i, j) \notin E$) **do**
    **if** $\tau_i = 0$ **then**
      update $\boldsymbol{\delta}_{N_i} = \{\delta_i^\alpha | \forall \alpha \in N_i\}$ with the closed-form update (11);
    **else if** $\tau_i \neq 0$ **then**
      update $\boldsymbol{\delta}_{N_i}$ and $\mathbf{w}_{N_i}$ with gradient descent (8) and(12), combined with line search;
    **end if**
  **end for**
**until** convergence
$\delta^* \leftarrow \delta$, $\mathbf{w}^* \leftarrow \mathbf{w}$, and evaluate $\mathrm{L}(\delta^*, \mathbf{w}^*)$ by (5);
*Remark.* GDD solves max-, sum- and mixed-inference by setting different values of weights $\{\tau_i\}$.

---

*where $\mu_i(x_i) \propto \exp\left[\frac{1}{w_i}\sum_{\alpha \in N_i} \delta_i^\alpha(x_i)\right]$ can be interpreted as a singleton belief on $x_i$, and $\mu_\alpha(x_\alpha)$ can be viewed as clique belief on $x_\alpha$, defined with a chain rule (assuming $x_\alpha = [x_1, \ldots, x_c]$), $\mu_\alpha(x_\alpha) = \prod_{i=1}^c \mu_\alpha(x_i | x_{i+1:c})$; $\mu_\alpha(x_i | x_{i+1:c}) = (Z_{i-1}(x_{i:c})/Z_i(x_{i+1:c}))^{1/w_i^\alpha}$, where $Z_i$ is the partial powered-sum up to $x_{1:i}$ on the clique, that is,*

$$Z_i(x_{i+1:c}) = \sum_{x_i}^{w_i^\alpha} \cdots \sum_{x_1}^{w_1^\alpha} \exp\left[\theta_\alpha(x_\alpha) - \sum_{i \in \alpha} \delta_i^\alpha(x_i)\right], \quad Z_0(x_\alpha) = \exp\left[\theta_\alpha(x_\alpha) - \sum_{i \in \alpha} \delta_i^\alpha(x_i)\right],$$

*where the summation order should be consistent with the global elimination order $\mathbf{o} = [x_1, \ldots, x_n]$.*

*(2) The gradients of $L(\delta, \mathbf{w})$ w.r.t. the weights $\{w_i, w_i^\alpha\}$ are marginal and conditional entropies defined on the beliefs $\{\mu_i, \mu_\alpha\}$, respectively,*

$$\frac{\partial L}{\partial w_i} = H(x_i; \mu_i), \qquad \frac{\partial L}{\partial w_i^\alpha} = H(x_i | x_{i+1:c}; \mu_\alpha) = -\sum_{x_\alpha} \mu_\alpha(x_\alpha) \log \mu_\alpha(x_i | x_{i+1:c}). \qquad (9)$$

*Therefore, the optimal weights should satisfy the following KKT condition*

$$w_i\big(H(x_i; \mu_i) - \bar{H}_i\big) = 0, \quad w_i^\alpha\big(H(x_i | x_{i+1:c}; \mu_\alpha) - \bar{H}_i\big) = 0, \quad \forall(i, \alpha) \qquad (10)$$

*where $\bar{H}_i = w_i H(x_i; \mu_i) + \sum_\alpha w_i^\alpha H(x_i | x_{i+1:c}; \mu_\alpha)$ is the (weighted) average entropy on node i.*

The proof details can be found in Section G of the supplement. The matching condition (8) enforces that $\mu = \{\mu_i, \mu_\alpha \mid \forall(i, \alpha)\}$ belong to the local consistency polytope $\mathbb{L}$ as defined in Theorem 4.2; similar moment matching results appear commonly in variational inference algorithms [e.g., 34]. [34] also derive a gradient of the weights, but it is based on the free energy form and is correct only after optimization; our form holds at any point, enabling efficient joint optimization of $\delta$ and $\mathbf{w}$.

## 5.2 Block Coordinate Descent

We derive a block coordinate descent method in Algorithm 1 to minimize our bound, in which we sweep through all the nodes $i$ and update each block $\boldsymbol{\delta}_{N_i} = \{\delta_i^\alpha(x_i) \mid \forall \alpha \in N_i\}$ and $\mathbf{w}_{N_i} = \{w_i, w_i^\alpha \mid \forall \alpha \in N_i\}$ with the neighborhood parameters fixed. Our algorithm applies two update types, depending on whether the variables have zero weight: (1) For nodes with $\tau_i = 0$ (corresponding to max nodes $i \in B$ in marginal MAP), we derive a closed-form coordinate descent rule for the associated shifting variables $\boldsymbol{\delta}_{N_i}$; these nodes do not require to optimize $\mathbf{w}_{N_i}$ since it is fixed to be zero. (2) For nodes with $\tau_i \neq 0$ (e.g., sum nodes $i \in A$ in marginal MAP), we lack a closed form update for $\boldsymbol{\delta}_{N_i}$ and $\mathbf{w}_{N_i}$, and optimize by local gradient descent combined with line search.

The lack of a closed form coordinate update for nodes $\tau_i \neq 0$ is mainly because the order of power sums with different weights cannot be exchanged. However, the gradient descent inner loop is still efficient, because each gradient evaluation only involves the local variables in clique $\alpha$.

**Closed-form Update.** For any node $i$ with $\tau_i = 0$ (i.e., max nodes $i \in B$ in marginal MAP), and its associated $\boldsymbol{\delta}_{N_i} = \{\delta_i^\alpha(x_i) \mid \forall \alpha \in N_i\}$, the following update gives a closed form solution for the zero (sub-)gradient equation in (8) (keeping the other $\{\delta_j^\alpha | j \neq i, \forall \alpha \in N_i\}$ fixed):

$$\delta_i^\alpha(x_i) \leftarrow \frac{|N_i|}{|N_i|+1}\gamma_i^\alpha(x_i) - \frac{1}{|N_i|+1}\sum_{\beta \in N_i \setminus \alpha}\gamma_i^\beta(x_i), \tag{11}$$

where $|N_i|$ is the number of neighborhood cliques, and $\gamma_i^\alpha(x_i) = \log \sum_{x_{\alpha \setminus i}}^{\mathbf{w}_{\setminus i}^\alpha} \exp\big[\theta_\alpha(x_\alpha) - \sum_{j \in \alpha \setminus i}\delta_j^\alpha(x_j)\big]$. Note that the update in (11) works regardless of the weights of nodes $\{\tau_j \mid \forall j \in \alpha,\ \forall \alpha \in N_i\}$ in the neighborhood cliques; when all the neighboring nodes also have zero weight ($\tau_j = 0$ for $\forall j \in \alpha,\ \forall \alpha \in N_i$), it is analogous to the "star" update of dual decomposition for MAP [31]. The detailed derivation is shown in Proposition H.2 in the supplement.

The update in (11) can be calculated with a cost of only $O(|N_i| \cdot d^{|\alpha|})$, where $d$ is the number of states of $x_i$, and $|\alpha|$ is the clique size, by computing and saving all the shared $\{\gamma_i^\alpha(x_i)\}$ before updating $\boldsymbol{\delta}_{N_i}$. Furthermore, the updates of $\boldsymbol{\delta}_{N_i}$ for different nodes $i$ are independent if they are not directly connected by some clique $\alpha$; this makes it easy to parallelize the coordinate descent process by partitioning the graph into independent sets, and parallelizing the updates within each set.

**Local Gradient Descent.** For nodes with $\tau_i \neq 0$ (or $i \in A$ in marginal MAP), there is no closed-form solution for $\{\delta_i^\alpha(x_i)\}$ and $\{w_i, w_i^\alpha\}$ to minimize the upper bound. However, because of the fully decomposed form, the gradient w.r.t. $\boldsymbol{\delta}_{N_i}$ and $\mathbf{w}_{N_i}$, (8)–(9), can be evaluated efficiently via local computation with $O(|N_i| \cdot d^{|\alpha|})$, and again can be parallelized between nonadjacent nodes. To handle the normalization constraint (6) on $\mathbf{w}_{N_i}$, we use an exponential gradient descent: let $w_i = \exp(v_i)/\big[\exp(v_i) + \sum_\alpha \exp(v_i^\alpha)\big]$ and $w_i^\alpha = \exp(v_i^\alpha)/\big[\exp(v_i) + \sum_\alpha \exp(v_i^\alpha)\big]$; taking the gradient w.r.t. $v_i$ and $v_i^\alpha$ and transforming back gives the following update

$$w_i \propto w_i \exp\big[-\eta w_i\big(H(x_i;\mu_i) - \bar{H}_i\big)\big], \quad w_i^\alpha \propto w_i^\alpha \exp\big[-\eta w_i^\alpha\big(H(x_i|x_{\mathrm{pa}_i^\alpha};\mu_\alpha) - \bar{H}_i\big)\big], \tag{12}$$

where $\eta$ is the step size and $\mathrm{pa}_i^\alpha = \{j \in \alpha \mid j \succ i\}$. In our implementation, we find that a few gradient steps (e.g., 5) with a backtracking line search using the Armijo rule works well in practice. Other more advanced optimization methods, such as L-BFGS and Newton's method are also applicable.

## 6 Experiments

In this section, we demonstrate our algorithm on a set of real-world graphical models from recent UAI inference challenges, including two diagnostic Bayesian networks with 203 and 359 variables and max domain sizes 7 and 6, respectively, and several MRFs for pedigree analysis with up to 1289 variables, max domain size of 7 and clique size 5.[7] We construct marginal MAP problems on these models by randomly selecting half of the variables to be max nodes, and the rest as sum nodes.

We implement several algorithms that optimize the same primal marginal MAP bound, including our GDD (Algorithm 1), the WMB algorithm in [16] with $ibound = 1$, which uses the same cliques and a fixed point heuristic for optimization, and an off-the-shelf L-BFGS implementation that directly optimizes our decomposed bound. For comparison, we also computed several related primal bounds, including standard mini-bucket [2] and elimination reordering [27, 38], limited to the same computational limits ($ibound = 1$). We also tried MAS [20] but found its bounds extremely loose.[8]

Decoding (finding a configuration $\hat{x}_B$) is more difficult in marginal MAP than in joint MAP. We use the same local decoding procedure that is standard in dual decomposition [31]. However, evaluating the objective $Q(\hat{x}_B)$ involves a potentially difficult sum over $x_A$, making it hard to score each decoding. For this reason, we evaluate the score of each decoding, but show the most recent decoding rather than the best (as is standard in MAP) to simulate behavior in practice.

Figure 2 and Figure 3 compare the convergence of the different algorithms, where we define the iteration of each algorithm to correspond to a full sweep over the graph, with the same order of time complexity: one iteration for GDD is defined in Algorithm 1; for WMB is a full forward and backward message pass, as in Algorithm 2 of [16]; and for L-BFGS is a joint quasi-Newton step on all variables. The elimination order that we use is obtained by a weighted-min-fill heuristic [1] constrained to eliminate the sum nodes first.

**Diagnostic Bayesian Networks.** Figure 2(a)-(b) shows that our GDD converges quickly and monotonically on both the networks, while WMB does not converge without proper damping; we

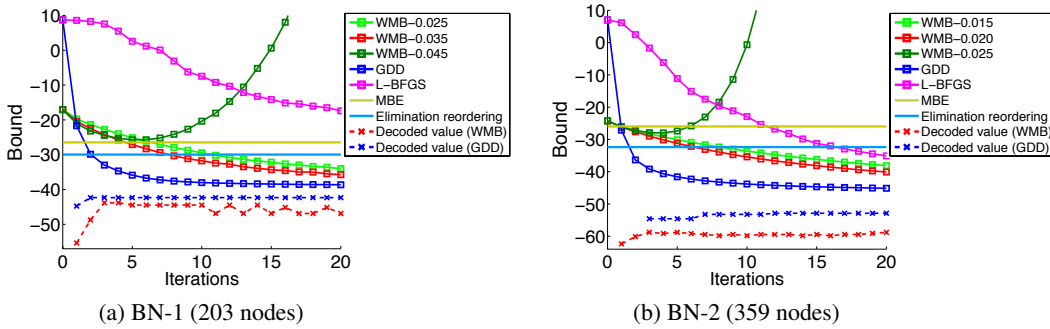

(a) BN-1 (203 nodes)  (b) BN-2 (359 nodes)

Figure 2: Marginal MAP results on BN-1 and BN-2 with $50\%$ randomly selected max-nodes (additional plots are in the supplement B). We plot the upper bounds of different algorithms across iterations; the objective function $Q(x_B)$ (2) of the decoded solutions $x_B$ are also shown (dashed lines). At the beginning, $Q(x_B)$ may equal to $-\infty$ because of zero probabiliy.

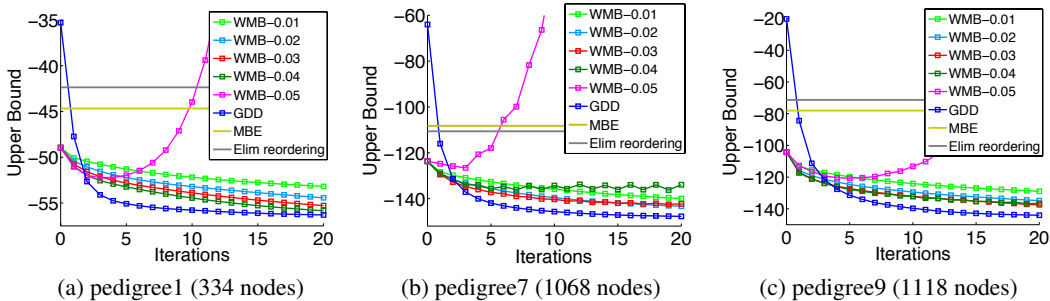

(a) pedigree1 (334 nodes)  (b) pedigree7 (1068 nodes)  (c) pedigree9 (1118 nodes)

Figure 3: Marginal MAP inference on three pedigree models (additional plots are in the supplement C). We randomly select half the nodes as max-nodes in these models. We tune the damping rate of WMB from 0.01 to 0.05.

experimented different damping ratios for WMB, and found that it is slower than GDD even with the best damping ratio found (e.g., in Figure 2(a), WMB works best with damping ratio 0.035 (WMB-0.035), but is still significantly slower than GDD). Our GDD also gives better decoded marginal MAP solution $x_B$ (obtained by rounding the singleton beliefs). Both WMB and our GDD provide a much tighter bound than the non-iterative mini-bucket elimination (MBE) [2] or reordered elimination [27, 38] methods.

**Genetic Pedigree Instances.** Figure 3 shows similar results on a set of pedigree instances. Again, GDD outperforms WMB even with the best possible damping, and out-performs the non-iterative bounds after only one iteration (pass through the graph).

## 7 Conclusion

In this work, we propose a new class of decomposition bounds for general powered-sum inference, which is capable of representing a large class of primal variational bounds but is much more computationally efficient. Unlike previous primal sum bounds, our bound decomposes into computations on small, local cliques, increasing efficiency and enabling parallel and monotonic optimization. We derive a block coordinate descent algorithm for optimizing our bound over both the cost-shifting parameters (reparameterization) and weights (fractional counting numbers), which generalizes dual decomposition and enjoy similar monotonic convergence property. Taking the advantage of its monotonic convergence, our new algorithm can be widely applied as a building block for improved heuristic construction in search, or more efficient learning algorithms.

**Acknowledgments**

This work is sponsored in part by NSF grants IIS-1065618 and IIS-1254071. Alexander Ihler is also funded in part by the United States Air Force under Contract No. FA8750-14-C-0011 under the DARPA PPAML program.

## Footnotes

[1]In some literature [e.g., 28], marginal MAP is simply called MAP, and the joint MAP task is called MPE.

[2]Despite the term "dual decomposition" used in MAP tasks, in this work we refer to decomposition bounds as "primal" bounds, since they can be viewed as directly bounding the result of variable elimination. This is in contrast to, for example, the linear programming relaxation of MAP, which bounds the result only after optimization.

[3]See an example on Ising model in Supplement A.

[4] The primal form derived in [5] (a geometric program) is computationally infeasible.

[5] See more details of this connection in Section F.3 of the supplement.

[6] While non-spanning subgraphs can be used in the primal TRW form, doing so leads to loose bounds; in contrast, our decomposition's terms consist of individual cliques.

[7]See http://graphmod.ics.uci.edu/uai08/Evaluation/Report/Benchmarks.

[8]The instances tested have many zero probabilities, which make finding lower bounds difficult; since MAS' bounds are symmetrized, this likely contributes to its upper bounds being loose.

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
