[Supplementary Material]

# Supplement: Decomposition Bounds for Marginal MAP

**Wei Ping**[*]       **Qiang Liu**[†]     **Alexander Ihler**[*]
[*]Computer Science, UC Irvine     [†]Computer Science, Dartmouth College
{wping,ihler}@ics.uci.edu   qliu@cs.dartmouth.edu

## A   Experiment on Ising grid

Our GDD directly optimizes a primal bound, and is thus guaranteed to be an upper bound of the partition function even before the algorithm converges, enabling a desirable "any-time" property. In contrast, typical implementations of tree reweighted (TRW) belief propagation optimize the dual free energy function [4], and are not guaranteed to be a bound before convergence. We illustrate this point using an experiment on a toy $5 \times 5$ Ising grid, with parameters generated by normal ditribution $N(0, 2)$ and half nodes selected as max-nodes for marginal MAP. Figure 1(a)-(b) shows the TRW free energy objective and GDD, WMB upper bounds across iterations; we can see that TRW does violate the upper bound property before convergence, while GDD and WMB always give valid upper bounds.

(a) sum-inference                    (b) marginal MAP

Figure 1: Sum-inference and marginal MAP results on a toy Ising model ($5 \times 5$ grid). Each iteration of the different algorithms corresponds to a full sweep over the graph. Note that the dual formulation (TRW) is not a bound until convergence; for example, at iteration 1, its objective function is below the true $\Phi$.

## B   More Results on Diagnostic Bayesian Networks

In addtion to the marginal MAP results on BN-1 and BN-2 in main text, we vary the percentage of max-nodes when generating the marginal MAP problems; the reported results in Figure 2(a)-(b) are the best bound obtained by the different algorithms with the first 20 iterations. In all cases, GDD's results are as good or better than WMB. WMB-0.5 (WMB with damping ratio $0.5$) appears to work well on sum-only and max-only (MAP) problems, i.e., when the percentage of max-nodes equals 0% and 100% respectively, but performs very poorly on intermediate settings. The far more heavily damped WMB-0.04 or WMB-0.02 work better on average, but have much slower convergence.

## C   More Results on Pedigree Linkage Analysis

We test our algorithm on additional 6 models of pedigree linkage analysis from the UAI08 inference challenge. We construct marginal MAP problems by randomly selected $50\%$ of nodes to be max-

(a) BN-1        (b) BN-2

Figure 2: More marginal MAP results (including sum-inference and MAP) on two diagnostic Bayesian networks. We report the best results obtained by GDD and WMB with 20 iterations in marginal MAP problems constructed by randomly selecting different percentages of max-nodes.

(a) pedigree13 (1077 nodes)    (b) pedigree18 (1184 nodes)    (c) pedigree19 (793 nodes)

(d) pedigree20 (437 nodes)    (e) pedigree23 (402 nodes)    (f) pedigree30 (1289 nodes)

Figure 3: Marginal MAP inference on additional pedigree linkage analysis models. We randomly selected $50\%$ of nodes as max-nodes in these models. We tune the damping rate of WMB from 0.01 to 0.06, but we omit WMB-0.06 in the plot if WMB-0.05 is already diverged.

nodes, and report all the results in Figure 3. We find that our algorithm consistently outperforms WMB with the best possible damping ratio.

## D  Extensions to Junction Graph

Our bound (5) in main text uses a standard "factor graph" representation in which the cost-shifts $\{\delta_i^\alpha\}$ are defined for each variable-factor pair $(i, \alpha)$, and are functions of single variables $x_i$. We can extend our bound to use more general shifting parameters using a junction graph representation; this allows us to exploit higher order clique structures, leading to better performance.

Let $(\mathcal{C}, \mathcal{S})$ be a junction graph of $p(x; \theta)$ where $\mathcal{C} = \{c \mid c \subset V\}$ is the set of clusters, and $\mathcal{S} = \{s = c_k \cap c_l \mid c_k, c_l \in \mathcal{C}\}$ is the set of separators. Assume $p(x; \theta)$ can be reparameterized into the form,

$$p(x; \theta) = \exp\left[\sum_{c \in \mathcal{C}} \theta_c(x_c) - \Phi(\theta)\right], \tag{1}$$

and the weighted log partition function is rewritten as $\Phi_{\boldsymbol{\tau}}(\theta) = \log \sum_x^{\boldsymbol{\tau}} \exp\left[\sum_{c \in \mathcal{C}} \theta_c(x_c)\right]$. Similar to the derivation of bound (5) in main text, we can apply Theorem 4.1, but with a set of more general cost-shifting variables $\delta_s^c$, defined on each adjacent separator-cluster pair $(s, c)$; this gives the more general upper bound,

$$\Phi_{\boldsymbol{\tau}}(\theta) \leq \sum_{s \in \mathcal{S}} \log \sum_{x_s}^{\mathbf{w}^s} \exp\left[\sum_{c \supseteq s} \delta_s^c(x_s)\right] + \sum_{c \in \mathcal{C}} \log \sum_{x_c}^{\mathbf{w}^c} \exp\left[\theta_c(x_c) - \sum_{s \subseteq c} \delta_s^c(x_s)\right], \qquad (2)$$

where we introduce the set of non-negative weights $\mathbf{w}^s = \{w_i^s \mid i \in s\}$ on each separator and $\mathbf{w}^c = \{w_i^c \mid i \in c\}$ on each cluster, which should satisfy $\sum_{s \in N_i^{se}} w_i^s + \sum_{c \in N_i^c} w_i^c = \tau_i$, where $N_i^{se} = \{s \mid i \in s\}$ are all the separators that include node $i$, and $N_i^c = \{c \mid i \in c\}$ are all the clusters that include node $i$. Obviously, our earlier bound (5) in main text can be viewed as a special case of (2) with a special junction graph whose separators consist of only single variables, that is, $\mathcal{S} = V$.

A block coordinate descent algorithm similar to Algorithm 1 can be derived to optimize the junction graph bound. In this case, we sweep through all the separators $s$ and perform block coordinate update on all $\{\delta_s^c | \forall c \supseteq s\}$ at each iteration. Similarly to Algorithm 1, we can derive a close form update for separators with all-zero weights (that is, $\tau_i = 0$, $\forall i \in s$, corresponding to $s \subseteq B$ in marginal MAP), and perform local gradient descent otherwise.

## E  Proof of Thereom 4.1

*Proof.* Note the Hölder's inequality is

$$\left[\sum_x \prod_j f_j(x)^{1/\xi_0}\right]^{\xi_0} \leq \prod_j \left[\sum_x f_j(x)^{1/\xi_j}\right]^{\xi_j},$$

where $\{f_j(x)\}$ are arbitrary positive functions, and $\{\xi_j\}$ are non-negative numbers that satisfy $\sum_j \xi_j = \xi_0$. Note we extend the inequality by defining power sum with $\xi_j = 0$ to equal the max operator. Our result follows by applying Hölder's inequality on each $x_i$ sequentially along the elimination order $[x_1, x_2, \cdots, x_n]$. $\qquad \square$

## F  Dual Representations

### F.1  Background

The log-partition function $\Phi(\theta)$ has the following variational (dual) form

$$\Phi(\theta) = \log \sum_x \exp(\theta(x)) = \max_{b \in \mathbb{M}(G)} \left\{\langle \theta, b \rangle + H(x; b)\right\}$$

where $\mathbb{M}(G)$ is the *marginal polytope* [5]. Then, for any scalar $\varepsilon > 0$ (including $\varepsilon \to 0^+$), we have

$$\Phi_\varepsilon(\theta) = \varepsilon \log \sum_x \exp(\frac{\theta(x)}{\varepsilon}) = \varepsilon \max_{b \in \mathbb{M}} \left\{\langle \frac{\theta}{\varepsilon}, b \rangle + H(x; b)\right\} = \max_{b \in \mathbb{M}} \left\{\langle \theta, b \rangle + \varepsilon H(x; b)\right\}.$$

As stated in [2, 3], we can further generalize the variational form of above scalar-weighted log partition function to the vector-weighted log partition function (3) in the main text,

$$\Phi_{\boldsymbol{\tau}}(\theta) = \log \sum_{x_n}^{\tau_n} \cdots \sum_{x_1}^{\tau_1} \exp(\theta(x)) = \max_{b \in \mathbb{M}(G)} \left\{\langle \theta, b \rangle + \sum_i \tau_i H(x_i | x_{i+1:n}; b)\right\}, \qquad (3)$$

where $H(x_i | x_{i+1:n}; b)$ is the conditional entropy on $b(x)$, and is defined as $H(x_i | x_{i+1:n}; b) = -\sum_x b(x) \log(b(x_i | x_{i+1:n}))$. See more details of its derivation in Theorem 4.1 within [2].

### F.2  Proof of Thereom 4.2

We will prove the following dual representation of our bound,

$$\min_\delta L(\delta, \mathbf{w}) = \max_{\mathbf{b} \in \mathbb{L}(G)} \left\{\langle \theta, b \rangle + \sum_{i \in V} w_i H(x_i; b_i) + \sum_{\alpha \in \mathcal{F}} \sum_{i \in \alpha} w_i^\alpha H(x_i | x_{\mathrm{pa}_i^\alpha}; b_\alpha)\right\}, \qquad (4)$$

where $\mathbb{L}(G) = \{\mathbf{b} \mid b_i(x_i) = \sum_{x_{\alpha \setminus i}} b_\alpha(x_\alpha), \; \sum_{x_i} b_i(x_i) = 1\}$ is the local consistency polytope, and $\mathrm{pa}_i^\alpha = \{j \in \alpha | j \succ i\}$. Thereom 4.2 follows directly from (4).

*Proof.* In our primal bound $L(\delta, \mathbf{w})$ (5) in main text, we let $\widetilde{\theta}_i(x_i) = \theta_i(x_i) + \sum_{\alpha \in N_i} \delta_i^\alpha(x_i)$ (we add dummy singleton $\theta_i(x_i) \equiv 0$), and $\widetilde{\theta}_\alpha(x_\alpha) = \theta_\alpha(x_\alpha) - \sum_{i \in \alpha} \delta_i^\alpha(x_i)$, then the bound can be rewritten as,

$$L(\widetilde{\theta}, \mathbf{w}) = \sum_{i \in V} \log \sum_{x_i}^{w_i} \exp\left[\widetilde{\theta}_i(x_i)\right] + \sum_{\alpha \in \mathcal{F}} \log \sum_{x_\alpha}^{\mathbf{w}^\alpha} \exp\left[\widetilde{\theta}_\alpha(x_\alpha)\right].$$

Note, for any assignment $x$, we have $\sum_i \widetilde{\theta}_i(x_i) + \sum_\alpha \widetilde{\theta}_\alpha(x_\alpha) = \sum_\alpha \theta_\alpha(x_\alpha)$.

By applying the dual form of the powered sum (3) on each node and clique respectively, we have

$$L(\widetilde{\theta}, \mathbf{w}) = \sum_{i \in V} \max_{b_i \in \mathbb{M}(G_i)} \left\{ \langle \widetilde{\theta}_i, b_i \rangle + w_i H(x_i; b_i) \right\} + \sum_{\alpha \in \mathcal{F}} \max_{b_\alpha \in \mathbb{M}(G_\alpha)} \left\{ \langle \widetilde{\theta}_\alpha, b_\alpha \rangle + \sum_{i \in \alpha} w_i^\alpha H(x_i | x_{\mathrm{pa}_i^\alpha}; b_\alpha) \right\},$$

where $\mathrm{pa}_i^\alpha$ is the set of variables in $\alpha$ that are summed out later than $i$, $\mathbb{M}(G_i)$ and $\mathbb{M}(G_\alpha)$ are the marginal polytopes on singleton node $i$ and clique $\alpha$ respectively, which enforce $\{b_i, b_\alpha\}$ to be properly normalized. We denote them jointly as $\widetilde{\mathbb{M}} = \{\mathbb{M}(G_i), \mathbb{M}(G_\alpha) \mid \forall i \in V, \alpha \in \mathcal{F}\}$, then

$$L(\widetilde{\theta}, \mathbf{w}) = \max_{\mathbf{b} \in \widetilde{\mathbb{M}}} \left\{ \langle \widetilde{\theta}, b \rangle + \sum_{i \in V} w_i H(x_i; b_i) + \sum_{\alpha \in \mathcal{F}} \sum_{i \in \alpha} w_i^\alpha H(x_i | x_{\mathrm{pa}_i^\alpha}; b_\alpha) \right\},$$

where $b_i, b_\alpha \in \widetilde{\mathbb{M}}$ are independently optimized.

Then, by tightening *reparameterization* $\widetilde{\theta} = \{\widetilde{\theta}_i, \widetilde{\theta}_\alpha\}$, we have

$$\min_{\widetilde{\theta}} L(\widetilde{\theta}, \mathbf{w}) = \max_{\mathbf{b} \in \widetilde{\mathbb{M}}} \min_{\widetilde{\theta}} \left\{ \langle \widetilde{\theta}, b \rangle + \sum_{i \in V} w_i H(x_i; b_i) + \sum_{\alpha \in \mathcal{F}} \sum_{i \in \alpha} w_i^\alpha H(x_i | x_{\mathrm{pa}_i^\alpha}; b_\alpha) \right\}$$

where the order of $\min$ and $\max$ are commuted according to the strong duality (it's convex with $\widetilde{\theta}$, and concave with $\mathbf{b}$).

The inner minimization $\min_{\widetilde{\theta}} \langle \widetilde{\theta}, b \rangle$ is a linear program, and it turns out can be solved analytically. To see this, given the relationship between $\widetilde{\theta}$ and $\theta$, we rewrite the linear program as

$$\min_{\widetilde{\theta}} \langle \widetilde{\theta}, b \rangle = \min_\delta \left\{ \langle \theta, b \rangle + \sum_{i \in V} \sum_{x_i} \sum_{\alpha \in N_i} \delta_i^\alpha(x_i) b_i(x_i) - \sum_{\alpha \in \mathcal{F}} \sum_{x_\alpha} \sum_{i \in \alpha} \delta_i^\alpha(x_i) b_\alpha(x_\alpha) \right\},$$

$$= \min_\delta \left\{ \langle \theta, b \rangle + \sum_{(i,\alpha)} \sum_{x_i} \delta_i^\alpha(x_i) \Big( b_i(x_i) - \sum_{x_{\alpha \setminus i}} b_\alpha(x_\alpha) \Big) \right\}.$$

Then, it is easy to observe that the linear program is either equal to $\langle \theta, b \rangle$ only if $b$ satisfy the marginalization constraint $\sum_{x_{\alpha \setminus i}} b_\alpha(x_\alpha) = b_i(x_i)$ for $\forall (i, \alpha)$, or it will become negative infinity. Considering the outer maximization, we have

$$\min_{\widetilde{\theta}} L(\widetilde{\theta}, \mathbf{w}) = \max_{\mathbf{b} \in \mathbb{L}(G)} \left\{ \langle \theta, b \rangle + \sum_{i \in V} w_i H(x_i; b_i) + \sum_{\alpha \in \mathcal{F}} \sum_{i \in \alpha} w_i^\alpha H(x_i | x_{\mathrm{pa}_i^\alpha}; b_\alpha) \right\},$$

where $\mathbb{L}(G)$ is the local consistency polytope that is obtained by enforcing both $\widetilde{\mathbb{M}}$ and the marginalization constraint. □

### F.3 Connection with Existing Free Energy Forms

Most variational forms are expresssed in the following linear combination of local entropies [6, 1],

$$\langle \theta, b \rangle + \sum_\beta c_\beta H(b_\beta), \tag{5}$$

where $\beta$ refers the region, $c_\beta$ refers the general counting number, $b_\beta(x_\beta)$ is the local belief.

We can rewrite our dual representations (4) as,

$$\langle\theta,b\rangle + \sum_{i\in V} w_i H(x_i; b_i) + \sum_{\alpha\in\mathcal{F}}\sum_{i\in\alpha} w_i^\alpha\big(H(x_i, x_{\mathrm{pa}_i^\alpha}\; ;\; b_\alpha) - H(x_{\mathrm{pa}_i^\alpha}\; ;\; b_\alpha)\big),$$

where $\mathrm{pa}_i^\alpha$ is the set of variables in $\alpha$ that rank later than $i$. Without loss of generality, assuming $x_\alpha = [x_1, \cdots, x_i, x_j, \cdots x_c]$, i.e. $x_i$ and $x_j$ are adjacent in the order, we can get

$$\langle\theta,b\rangle + \sum_{i\in V} w_i H(x_i; b_i) + \sum_{\alpha\in\mathcal{F}}\left\{ w_1^\alpha H(x_\alpha; b_\alpha) + \sum_{[i,j]\sqsubseteq\alpha} (w_j^\alpha - w_i^\alpha) H(x_{\mathrm{pa}_i^\alpha}\; ;\; b_{\mathrm{pa}_i^\alpha})\right\} \quad (6)$$

where belief $b_{\mathrm{pa}_i^\alpha}$ is defined by $b_{\mathrm{pa}_i^\alpha}(x_{\mathrm{pa}_i^\alpha}) = \sum_{x_{\alpha\backslash\mathrm{pa}_i^\alpha}} b_\alpha(x_\alpha)$.

One can view (6) in terms of (5), by selecting the region $\beta\in\{i\in V\}\cup\{\alpha\in\mathcal{F}\}\cup\{\mathrm{pa}_i^\alpha\,|\,\forall(i,\alpha)\}$; some counting numbers $c_\beta$ will be the differences of weights $w_j^\alpha - w_i^\alpha$.

### F.4 Matching Our Bound to WMB

After the weights are optimized, our GDD bound matches to WMB bound with optimum weights. A simple weight initialization method matches our bound to WMB with uniform weights on each mini-bucket, which often gives satisfactory result; a similar procedure can be used to match the bound with more general weights as in Section D. We first set $w_i = 0$ for all nodes $i$. We then visit the nodes $x_i$ along the elimination order $\mathbf{o} = [x_1, x_2, \cdots, x_n]$, and divide $x_i$'s neighborhood cliques $N_i = \{\alpha|\alpha\ni i\}$ into two groups: (1) the *children cliques* in which all $x_{\alpha\backslash i}$ have already been eliminated, that is, $N_i^{ch} = \{\alpha\,|\,\forall j\in\alpha\backslash i,\ j\prec i\text{ in }\mathbf{o}\}$; (2) the other, *parent cliques* $N_i^{pa} = \{\alpha\,|\,\exists j\in\alpha\backslash i,\ j\succ i\text{ in }\mathbf{o}\}$. We set $w_i^\alpha = 0$ for all the children cliques ($\alpha\in N_i^{ch}$), and uniformly split the weights, that is, $w_i^\alpha = \tau_i/|N_i^{pa}|$, across the parent cliques.

## G  Proof of Therom 5.1

*Proof.* For each $\delta_i^\alpha(x_i)$, the involved terms in $L(\delta, \mathbf{w})$ are $L_i^\alpha(\delta) = \Phi_{w_i}(\delta) + \Phi_{\mathbf{w}^\alpha}(\delta)$, where

$$\Phi_{w_i}(\delta) = \log\sum_{x_i}^{w_i}\exp\Big[\sum_{\alpha\in N_i}\delta_i^\alpha(x_i)\Big], \quad \Phi_{\mathbf{w}^\alpha}(\delta) = \log\sum_{x_\alpha}^{\mathbf{w}^\alpha}\exp\Big[\theta_\alpha(x_\alpha) - \sum_{i\in\alpha}\delta_i^\alpha(x_i)\Big].$$

Our result follows by showing that

$$\frac{\partial\Phi_{w_i}(\delta)}{\partial\delta_i^\alpha(x_i)} = \mu_i(x_i) \qquad \text{and} \qquad \frac{\partial\Phi_{w_i}(\delta)}{w_i} = H(x_i; \mu_i),$$

$$\frac{\partial\Phi_{\mathbf{w}^\alpha}(\delta)}{\partial\delta_i^\alpha(x_i)} = -\sum_{x_{\alpha\backslash i}}\mu_\alpha(x_\alpha) \quad \text{and} \quad \frac{\partial\Phi_{\mathbf{w}^\alpha}(\delta)}{\partial w_i^\alpha} = H(x_i|x_{i+1:c}; \mu_\alpha).$$

The gradient of $\Phi_{w_i}(\delta)$ is straightforward to calculate,

$$\frac{\partial\Phi_{w_i}}{\partial\delta_i^\alpha(x_i)} = \frac{\partial}{\partial\delta_i^\alpha(x_i)}\Big(w_i\log\sum_{x_i}\exp\Big[\frac{\sum_{\alpha\in N_i}\delta_i^\alpha(x_i)}{w_i}\Big]\Big) = \frac{\exp\Big[\frac{\sum_{\alpha\in N_i}\delta_i^\alpha(x_i)}{w_i}\Big]}{Z_{w_i}} = \mu_i(x_i),$$

where $Z_{w_i} = \sum_{x_i}\exp\Big[\frac{\sum_{\alpha\in N_i}\delta_i^\alpha(x_i)}{w_i}\Big]$, and

$$\frac{\partial\Phi_{w_i}}{\partial w_i} = \log Z_{w_i} + w_i\cdot\frac{1}{Z_{w_i}}\cdot\sum_{x_i}\Big\{\exp\Big[\frac{\sum_{\alpha\in N_i}\delta_i^\alpha(x_i)}{w_i}\Big]\cdot\frac{\sum_{\alpha\in N_i}\delta_i^\alpha(x_i)}{-w_i^2}\Big\}$$

$$= \log Z_{w_i} - \sum_{x_i}\Big\{\mu_i(x_i)\cdot\frac{\sum_{\alpha\in N_i}\delta_i^\alpha(x_i)}{w_i}\Big\}$$

$$= -\sum_{x_i}\Big\{\mu_i(x_i)\cdot\Big[\frac{\sum_{\alpha\in N_i}\delta_i^\alpha(x_i)}{w_i} - \log Z_{w_i}\Big]\Big\} = H(x_i; \mu_i).$$

The gradient of $\Phi_{\mathbf{w}^\alpha}(\delta)$ is more involved; see Proposition I.1 for a detailed derivation.  $\square$

## H  Derivations of Closed-form Update

We first derive the closed-form update rule for $\delta_i^\alpha(x_i)$ in Proposition H.1. We derive the closed-form update rule for the block $\boldsymbol{\delta}_{N_i} = \{\delta_i^\alpha(x_i) \mid \forall \alpha \in N_i\}$ in Proposition H.2.

**Proposition H.1.** *Given max node $i$ in marginal MAP (i.e., $\tau_i = 0$) and one clique $\alpha \ni i$ (i.e. $i \in \alpha$), keeping all $\delta$ fixed except $\delta_i^\alpha(x_i)$, there is a closed-form update rule,*

$$\delta_i^\alpha(x_i) \leftarrow \frac{1}{2} \log \sum_{x_{\alpha \backslash i}}^{w_{\backslash i}^\alpha} \exp\left[\theta_\alpha(x_\alpha) - \sum_{j \in \alpha \backslash i} \delta_j^\alpha(x_j)\right] - \frac{1}{2} \sum_{\beta \in N_i \backslash \alpha} \delta_i^\beta(x_i), \tag{7}$$

*where $x_{\alpha \backslash i} = \{x_j : j \in \alpha, j \neq i\}$, $w_{\backslash i}^\alpha = \{w_j^\alpha : j \in \alpha, j \neq i\}$, and $N_i = \{\alpha \mid \alpha \ni i\}$ is the set of all clique factors in the neighborhood of node $i$. Futhermore, this update will monotonically decrease the upper bound.*

*Proof.* The terms within the bound $L(\delta, \mathbf{w})$ that depend on $\delta_i^\alpha(x_i)$ are,

$$\max_{x_i}\left[\sum_{\alpha \in N_i} \delta_i^\alpha(x_i)\right] + \max_{x_i} \log \sum_{x_{\alpha \backslash i}}^{w_{\backslash i}^\alpha} \exp\left[\theta_\alpha(x_\alpha) - \sum_{i \in \alpha} \delta_i^\alpha(x_i)\right] \tag{8}$$

The sub-gradient of (8) w.r.t. $\delta_i^\alpha(x_i)$ equal to zero if and only if,

$$x_i^* = \operatorname*{argmax}_{x_i}\left[\sum_{\alpha \in N_i} \delta_i^\alpha(x_i)\right] = \operatorname*{argmax}_{x_i} \log \sum_{x_{\alpha \backslash i}}^{w_{\backslash i}^\alpha} \exp\left[\theta_\alpha(x_\alpha) - \sum_{i \in \alpha} \delta_i^\alpha(x_i)\right],$$

which is "argmax" matching. One sufficient condition of this matching is,

$$\sum_{\alpha \in N_i} \delta_i^\alpha(x_i) = \log \sum_{x_{\alpha \backslash i}}^{w_{\backslash i}^\alpha} \exp\left[\theta_\alpha(x_\alpha) - \sum_{i \in \alpha} \delta_i^\alpha(x_i)\right]$$

which impllies matching of "pseudo marginals". Then, one can pull $\delta_i^\alpha(x_i)$ outside from the operator $\log \sum_{x_{\alpha \backslash i}}^{w_{\backslash i}^\alpha} \exp$, and get the closed-form equation

$$\delta_i^\alpha(x_i) = \frac{1}{2} \log \sum_{x_{\alpha \backslash i}}^{w_{\backslash i}^\alpha} \exp\left[\theta_\alpha(x_\alpha) - \sum_{j \in \alpha \backslash i} \delta_j^\alpha(x_j)\right] - \frac{1}{2} \sum_{\beta \in N_i \backslash \alpha} \delta_i^\beta(x_i).$$

To prove monotonicity, we substitute above update equation of $\delta_i^\alpha(x_i)$ into (8); then we get,

$$\max_{x_i}\left\{\sum_{\beta \in N_i \backslash \alpha} \delta_i^\beta(x_i) + \log \sum_{x_{\alpha \backslash i}}^{w_{\backslash i}^\alpha} \exp\left[\theta_\alpha(x_\alpha) - \sum_{j \in \alpha \backslash i} \delta_j^\alpha(x_j)\right]\right\}. \tag{9}$$

Clearly, (9) $\leq$ (8) by using the fact that $\max_x[f(x) + g(x)] \leq \max_x f(x) + \max_x g(x)$. $\square$

**Proposition H.2.** *Given node $i \in B$ (i.e., a max node) and all neighborhood cliques $N_i = \{\alpha \mid \alpha \ni i\}$, we can jointly optimize $\boldsymbol{\delta}_{N_i} = \{\delta_i^\alpha(x_i) \mid \forall \alpha \in N_i\}$ in closed-form by keeping the other $\{\delta_j^\alpha \mid j \neq i, \forall \alpha \in N_i\}$ fixed. The update rule is,*

$$\delta_i^\alpha(x_i) \leftarrow \frac{|N_i|}{|N_i| + 1} \gamma_i^\alpha(x_i) - \frac{1}{|N_i| + 1} \sum_{\beta \in N_i \backslash \alpha} \gamma_i^\beta(x_i), \tag{10}$$

*where $|N_i|$ is the number of neighborhood cliques, and $\{\gamma_i^\alpha(x_i) \mid \forall \alpha \in N_i\}$ are defined by*

$$\gamma_i^\alpha(x_i) = \log \sum_{x_{\alpha \backslash i}}^{\mathbf{w}_{\backslash i}^\alpha} \exp\left[\theta_\alpha(x_\alpha) - \sum_{j \in \alpha \backslash i} \delta_j^\alpha(x_j)\right]. \tag{11}$$

*Futhermore, this upate will monotonically decrease the upper bound.*

*Proof.* For $\forall \alpha \in N_i$, we have closed-form solutions for $\delta_i^\alpha(x_i)$ as Proposition H.1. We rewrite it as,

$$\forall \alpha \in N_i, \quad 2\delta_i^\alpha(x_i) + \sum_{\beta \in N_i \setminus \alpha} \delta_i^\beta(x_i) = \log \sum_{x_{\alpha \setminus i}}^{w_{\setminus i}^\alpha} \exp \left[ \theta_\alpha(x_\alpha) - \sum_{j \in \alpha \setminus i} \delta_j^\alpha(x_j) \right]. \tag{12}$$

Note, for $\forall \alpha, \beta \in N_i$, there is a linear relationship between $\delta_i^\alpha(x_i)$ and $\delta_i^\beta(x_i)$.

We denote column vector $\boldsymbol{\gamma}_i(x_i)$ filled $\alpha$-th element with

$$\gamma_i^\alpha(x_i) = \log \sum_{x_{\alpha \setminus i}}^{w_{\setminus i}^\alpha} \exp \left[ \theta_\alpha(x_\alpha) - \sum_{j \in \alpha \setminus i} \delta_j^\alpha(x_j) \right].$$

We also frame all $\{\delta_i^\alpha(x_i) \mid \alpha \in N_i\}$ into a column vector $\boldsymbol{\delta}_{N_i}(x_i)$, and denote $|N_i| \times |N_i|$ matrix A

$$A = \begin{pmatrix} 2 & 1 & \cdots & 1 \\ 1 & 2 & \cdots & 1 \\ \vdots & \vdots & \ddots & \vdots \\ 1 & 1 & \cdots & 2 \end{pmatrix}, \text{ and note } A^{-1} = \begin{pmatrix} \frac{|N_i|}{|N_i|+1} & -\frac{1}{|N_i|+1} & \cdots & -\frac{1}{|N_i|+1} \\ -\frac{1}{|N_i|+1} & \frac{|N_i|}{|N_i|+1} & \cdots & -\frac{1}{|N_i|+1} \\ \vdots & \vdots & \ddots & \vdots \\ -\frac{1}{|N_i|+1} & -\frac{1}{|N_i|+1} & \cdots & \frac{|N_i|}{|N_i|+1} \end{pmatrix}.$$

It is easy to verify $A\boldsymbol{\delta}_{N_i}(x_i) = \boldsymbol{\gamma}_i(x_i)$. from (12). Since $A$ is invertible, one can solve

$$\boldsymbol{\delta}_{N_i}(x_i) = A^{-1}\boldsymbol{\gamma}_i(x_i).$$

Then, one can read out the closed-form update rule (10). The monotonicity holds directly by noticing that the update rule (10) are solutions which jointly satisfy equation (7). □

# I    Derivations of Gradient

**Proposition I.1.** *Given a weight vector* $\mathbf{w}^\alpha = [w_1^\alpha, \cdots, w_i^\alpha, \cdots, w_c^\alpha]$ *associated with variables* $x_\alpha = \{x_1, \cdots, x_i, \cdots, x_c\}$ *on clique* $\alpha$, *where* $c = |\alpha|$ *the power sum over clique* $\alpha$ *is,*

$$\Phi_{\mathbf{w}^\alpha}(\delta) = \log \sum_{x_\alpha}^{\mathbf{w}^\alpha} \exp \left[ \theta_\alpha(x_\alpha) - \sum_{i \in \alpha} \delta_i^\alpha(x_i) \right] = \log \sum_{x_c}^{w_c^\alpha} \cdots \sum_{x_i}^{w_i^\alpha} \cdots \sum_{x_1}^{w_1^\alpha} \exp \left[ \theta_\alpha(x_\alpha) - \sum_{i \in \alpha} \delta_i^\alpha(x_i) \right].$$

*We recursively denote* $Z_i$ *as the partial power sum up to* $x_{1:i}$,

$$Z_0(x_\alpha) = \exp \left[ \theta_\alpha(x_\alpha) - \sum_{i \in \alpha} \delta_i^\alpha(x_i) \right] \quad and \quad Z_i(x_{i+1:c}) = \sum_{x_i}^{w_i^\alpha} Z_{i-1}(x_{i:c}), \tag{13}$$

*thus* $\log Z_c = \Phi_{\mathbf{w}^\alpha}$. *We also denote the "pseudo marginal" (or, belief) on* $x_\alpha$,

$$\mu_\alpha(x_\alpha) = \prod_{i=1}^c \mu_\alpha(x_i|x_{i+1:c}); \quad \mu_\alpha(x_i|x_{i+1:c}) = \left( \frac{Z_{i-1}(x_{i:c})}{Z_i(x_{i+1:c})} \right)^{1/w_i^\alpha},$$

*and it is easy to verify that* $\mu_\alpha(x_i|x_{i+1:c})$ *and* $\mu_\alpha(x_\alpha)$ *are normalized.*

*Then, the derivative of* $\Phi_{\mathbf{w}^\alpha}$ *w.r.t.* $\delta_i^\alpha(x_i)$ *can be written by beliefs,*

$$\frac{\partial \Phi_{\mathbf{w}^\alpha}}{\partial \delta_i^\alpha(x_i)} = -\mu_\alpha(x_i) = -\sum_{x_{\alpha \setminus i}} \mu_\alpha(x_\alpha) = -\sum_{x_c} \cdots \sum_{x_{i+1}} \prod_{j=i}^c \mu_\alpha(x_j|x_{j+1:c}) \tag{14}$$

*In addition, the derivative of* $\Phi_{\mathbf{w}^\alpha}$ *w.r.t.* $w_i^\alpha$ *is the conditional entropy,*

$$\frac{\partial \Phi_{\mathbf{w}^\alpha}}{\partial w_i^\alpha} = H(x_i|x_{i+1:c}; \mu_\alpha(x_\alpha)) = -\sum_{x_\alpha} \mu_\alpha(x_\alpha) \log \mu_\alpha(x_i|x_{i+1:c}) \tag{15}$$

*Proof.*

Denote the reparameterization on clique $\alpha$ as $\widetilde{\theta}_\alpha(x_\alpha) = \theta_\alpha(x_\alpha) - \sum_{i \in \alpha} \delta_i^\alpha(x_i)$.

From the recursive definition of $Z_i(x_{i+1:c})$ (13), we have the following recursive rule for gradient,

$$
\begin{aligned}
\frac{\partial \log Z_i(x_{i+1:c})}{\partial \widetilde{\theta}_\alpha(x_\alpha)} &= \frac{\partial}{\partial \widetilde{\theta}_\alpha(x_\alpha)} \Big( w_i^\alpha \log \sum_{x_i} \big[ Z_{i-1}(x_{i:c}) \big]^{1/w_i^\alpha} \Big) \\
&= w_i^\alpha \cdot \frac{\frac{1}{w_i^\alpha} \cdot Z_{i-1}(x_{i:c})^{\frac{1}{w_i^\alpha}}}{\sum_{x_i} \big[ Z_{i-1}(x_{i:c}) \big]^{\frac{1}{w_c^\alpha}}} \cdot Z_{i-1}(x_{i:c})^{-1} \cdot \frac{\partial Z_{i-1}(x_{i:c})}{\partial \widetilde{\theta}_\alpha(x_\alpha)} \\
&= \frac{Z_{i-1}(x_{i:c})^{\frac{1}{w_i^\alpha}}}{\sum_{x_i} \big[ Z_{i-1}(x_{i:c}) \big]^{\frac{1}{w_c^\alpha}}} \cdot \frac{\partial \log Z_{i-1}(x_{i:c})}{\partial \widetilde{\theta}_\alpha(x_\alpha)} \\
&= \mu_\alpha(x_i | x_{i+1:c}) \cdot \frac{\partial \log Z_{i-1}(x_{i:c})}{\partial \widetilde{\theta}_\alpha(x_\alpha)}.
\end{aligned}
\tag{16}
$$

It should be noted, implicitly, $x_{i+1:c}$ within $\widetilde{\theta}_\alpha(x_\alpha)$ should take the same value as $x_{i+1:c}$ in $\log Z_i(x_{i+1:c})$, otherwise, the derivative will equal 0.

As a result, we can calculate the derivatives of $\Phi_{\mathbf{w}^\alpha}(\widetilde{\theta}_\alpha)$ w.r.t. $\widetilde{\theta}_\alpha(x_\alpha)$ recursively as,

$$
\frac{\partial \Phi_{\mathbf{w}^\alpha}}{\partial \widetilde{\theta}_\alpha(x_\alpha)} = \frac{\partial \log Z_c}{\partial \widetilde{\theta}_\alpha(x_\alpha)} = \mu_\alpha(x_c) \cdot \frac{\partial \log Z_{c-1}(x_c)}{\partial \widetilde{\theta}_\alpha(x_\alpha)} = \cdots = \prod_{i=1}^c \mu_\alpha(x_i | x_{i+1:c}) = \mu_\alpha(x_\alpha).
\tag{17}
$$

By the chain rule,

$$
\frac{\partial \Phi_{\mathbf{w}^\alpha}}{\partial \delta_i^\alpha(x_i)} = \sum_{x_{\alpha \setminus i}} \frac{\partial \Phi_{\mathbf{w}^\alpha}}{\partial \widetilde{\theta}_\alpha(x_i, x_{\alpha \setminus i})} \cdot \frac{\partial \widetilde{\theta}_\alpha(x_i, x_{\alpha \setminus i})}{\partial \delta_i^\alpha(x_i)} = -\sum_{x_{\alpha \setminus i}} \mu_\alpha(x_\alpha),
$$

then (14) has been proved.

Applying the variational form of powered-sum (3) to $\Phi_{\mathbf{w}^\alpha}$, we have

$$
\Phi_{\mathbf{w}^\alpha}(\widetilde{\theta}_\alpha) = \max_{b_\alpha \in \mathbb{M}_\alpha(G)} \big\{ \langle \widetilde{\theta}_\alpha, b_\alpha \rangle + \sum_i w_i^\alpha H(x_i | x_{i+1:n}; b_\alpha) \big\}.
$$

According to Danskin's theorem, the derivative $\frac{\partial \Phi_{\mathbf{w}^\alpha}}{\partial \widetilde{\theta}_\alpha(x_\alpha)} = b_\alpha^*(x_\alpha)$, which is the optimum of RHS. Combined with (17), we have $b_\alpha^* = \mu_\alpha$ immediately, and the derivative w.r.t. $w_i^\alpha$ is,

$$
\frac{\partial \Phi_{\mathbf{w}^\alpha}}{\partial w_i^\alpha} = H(x_i | x_{i+1:c}; \mu_\alpha(x_\alpha)),
$$

then (15) has been proved.

$\square$