[Reviews · NeurIPS 2015]

Submitted by Assigned_Reviewer_1

The authors present a dual decomposition method for marginal MAP problems that makes use of Holder's inequality.

This derivation stands apart from previous work in that it requires a fixed elimination ordering for the bounds to hold - the order of operations is not commutative.

Still, the authors show that the resulting optimization problem has a similar form to known results in the literature on approximate variational inference.

The paper is well-written and easy to read.

General comments:

-pg. 2, "the resulting optimization is only guaranteed to be a bound at convergence..." I'm not sure that this is true.

I know for a fact it isn't true for TRW max-product despite some published statements to the contrary (though it does require a slightly different reparameterization vantage point).

You should take another look at "Convergent message passing algorithms - a unifying view" by Meltzer et al.

I think that the argument for the max-product case can be applied to the sum-product case as well.

The authors made several comments on this point (which I admittedly didn't fully understand).

To clarify, I was thinking of techniques based on reparameterization and pushing max through a product to get an inequality.

In these situations (much like yours), you have a bound for any choice of messages even when the consistency conditions are not satisfied.

-The reparameterization approach described here is similar to "Message-passing algorithms: reparameterizations and splittings" by Ruozzi et al.

You should probably cite it.

Typos:

-pg. 2, extra space in "Section

2" -throughout:

"message passing algorithm" -> "message-passing algorithm" -throughout:

I don't think, grammatically, that citations should be used as nouns.

"[8] showed" -> "Hazan et al. [8] showed"
Summary: The authors present a dual decomposition method for marginal MAP problems that makes use of Holder's inequality.

The paper is well-written and easy to read.

Submitted by Assigned_Reviewer_2

Marginal MAP is a very difficult inference problem on which there has been relatively little research efforts (comparing to MAP inference and marginal inference). This paper develops a new decomposition-based upper bound to the marginal MAP problem, as well as monotonic optimization methods for the bound. The propose method generalizes the popular dual decomposition framework for MAP inference. Experimental results on real data have verified the effectiveness of the proposed approach.

A key ingredient of the framework is the power sum operator, featuring a temperature parameter relating MAP inference and marginal inference. The idea is to apply different temperature to different variables: low temperature for those we want to maximize over, and high temperatures for those we want to marginalize out. A bunch of "splitting weights" (dual variables) help to define an upper-bound to the marginal inference problem. The splitting weights are altered by "cost-shifting" variables which are essentially messages passed around the graph to minimize the upper bound. The framework includes MAP inference dual decomposition as a special case.

A block coordinate descent method is developed for the decomposed bound. For those variables with zero temperature (max nodes), the update rule resembles that of the MAP inference dual decomposition case. For those variables with non-zero temperatures (sum nodes), there isn't such a simple update rule. So the authors had to resorted to gradient descent and line search in those dimensions.

Experiments on standard benchmarks have shown that the proposed method significantly outperforms baselines.

Summary: The framework generalizes dual decomposition to marginal MAP. It seems to have solid theoretical and practical contributions to the field.

Submitted by Assigned_Reviewer_3

The paper works on the problem of marginal MAP inference in graphical models, which first marginalise a set of variables and then find the MAP configuration of the rest variables. It derives a decomposable bound on the problem, which can be considered as a generalisation of dual decomposition for MAP inference and TRW for marginal inference. Based on the bound, the authors propose an algorithm for marginal MAP inference (Algorithm 1).

* The paper is well written and very easy to follow. * The approach is well motivated and technically correct. * The result has both theoretical and algorithmic significance:

- It generalises and unifies the bounds derived in existing research works.

- It leads to an algorithm which is faster and more reliable than previous methods. * The marginal MAP inference have an important role in learning latent variable models. This work is likely to have impact on latent variable problems. * The two central ideas of this paper: i) using powered-sum to unify max and sum operations and ii) deriving upper bound of the powered-sum from Holder's inequality have existed in existing works: [16] and [17] respectively. Thus, the idea of this paper is probably not very original but it is still an important incremental work on [16] and [17].
Summary: The paper derives a decomposable upper bound for the problem of MAP inference. It is well written, interesting and technically sound and thus should be considered to be accepted by NIPS.

Author Feedback
Author rebuttal: We thank all the reviewers for their helpful suggestions. In particular, thanks to "heavy" reviewers AR1, 3, and 4, for their detailed comments.

Reviewer 2:
Thanks for the reference. We will read the paper and make a connection.

Reviewer 4:
--Related Work: "On the other hand, the dual, or free energy... the resulting optimization is only guaranteed to be a bound at convergence..."
Our argument holds for the free energy form, in which the upper bound is a constrained maximization; if not yet maximized, it may not be a bound (and typically, the local consistency constraints are also not enforced until convergence, making its relationship to the true value unclear). However, some methods obtain a primal bound by re-taking a dual of the free energy. For example, Meltzer et al. propose a family of re-dualized bounds, so as to provide a bound at any point.

To be more specific, we replicate Eq. (10) and its preceding inequality from Meltzer et al., and abbreviate '\alpha' as 'a':
max_{q} F = max_q \sum_a ( < theta_a, q_a > + c_a H_a ) , (I)
<= \sum_a max_{q_a} ( < theta_a, q_a > + c_a H_a ) , (II)
= \sum_a c_a ln Z_{theta_a} . (III)
(I) is the "dual" (free energy) form that most methods optimize (e.g. Wainright et al. 2005, Hazan et al. 2008);
(II) is a relaxation of that form, which upper bounds the optimum;
(III) is the dual form of (II), giving a primal, "decomposition-like" bound. Often, the free energy beliefs can be re-transformed into such a decomposition bound. But, this one requires that c_a be all positive, which is not always easily satisfied and hard to apply to marginal MAP.

-- Thanks for pointing to Ruozzi et al. TIT13; we will cite it in full-length version. We also appreciate you pointing out the typos, and will revise accordingly.

Reviewer 5:
Thanks for your feedback. Agreed, but "Inferning" was not coined by us; it was used as a workshop name in ICML'12 and ICML'13. Perhaps we can refer to it differently.